# Development of the Architecture and Reconfiguration Methods for the Smart, Self-Reconfigurable Manufacturing System

Sangil Lee and Kwangyeol Ryu *

Department of Industrial Engineering, Pusan National University, 2, Busandaehak-ro 63beon-gil, Geumjeong-gu, Jangjeon-dong, Busan 46241, Korea; lscall@pusan.ac.kr
* Correspondence: kyryu@pusan.ac.kr; Tel.: +82-51-510-2473

**Abstract:** Over recent decades, the demand for smarter and more intelligent manufacturing systems has increased in order to meet the growing requirements of customers. Manufacturing systems are termed as smart manufacturing systems (SMSs); these systems are capable of fully integrated autonomous operation. Specifically, the concept of autonomous systems and functions has been adopted for next generation manufacturing systems (NGMSs). Among these NGMSs, the fractal manufacturing system (FrMS) exhibits several characteristics that are similar to those of SMSs. Therefore, in this paper, a smart, self-reconfigurable manufacturing system (SSrMS) based on the FrMS is proposed. The proposed SSrMS architecture was designed for realizing self-reconfiguration functions based on the FrMS concept. SSrMS exhibits a fractal structure, which enables the distribution of control features; this also constitutes the fundamental basis of autonomous operation and reconfiguration between each fractal. SSrMS architecture includes the use of big data, digital facilities, and simulations. Furthermore, we introduce three reconfiguration methods to conduct system reconfiguration, which are a goal decision model, a negotiation model, and a sustainability assessment method. The goal decision model was developed to determine a goal of each fractal to achieve the system's goal. In other words, each fractal can decide a goal to achieve the system's goal, such as maximizing productivity or profit, or minimizing cost, and others. The negotiation model was adopted to perform partial process optimization by reassigning tasks and resources between the fractals, based on the goal of coping with the changes in the system's condition. The sustainability assessment method was designed to simultaneously evaluate sustainability with respect to the system's goals. The proposed architecture of SSrMS with goal decision model, negotiation model, and sustainability assessment method has the features of self-optimization, self-organization, and self-reconfiguration in order to achieve fully autonomous operations for the manufacturing system. The proposed architecture including three methods are expected to provide a fundamental study of the autonomous operations. The main findings of in this study is the development of a new architecture for fully autonomous operations of the smart manufacturing system with reconfiguration methods of goal-oriented manufacturing processes.

**Keywords:** fractal manufacturing system (FrMS); self-reconfigurable manufacturing system; smart manufacturing; Industry 4.0; goal decision model; negotiation model; sustainability assessment method



## 1. Introduction

Since the advent of Industry 4.0 as a new paradigm, the manufacturing industry has evolved significantly. Furthermore, the paradigm of a new manufacturing system shifted rapidly due to the increasing demands of mass personalization, sustainable production, sustainable products, etc. Hence, a variety of smart manufacturing concepts and architectures have been suggested. Kusiak [1] suggested the latest term of smart manufacturing systems (SMSs), which includes six essential pillars: materials, data, predictive engineering, sustainability, resource sharing and networking, and manufacturing technology and

processes [2,3]. SMSs employ similar technologies, such as information and communication technology (ICT), industrial internet of things (IIoT), artificial intelligence (AI), big data, sensor data, etc. These technologies have led to the smartization of manufacturing and highly connected resources throughout the process [4]. Qu et al. [2] described fully integrated autonomous functions for smart manufacturing, as follows:

- Integration, organization, and allocation for advanced information and communication technologies [5];
- Self-optimizing processes [6], data-driven decision making [7], self-regulation, and self-organization for adaptive and intelligent manufacturing control [8];
- Predictive situational awareness [9], statistical process monitoring and prognostics [10], predictive maintenance [10], and predictive manufacturing situations [11] for operation reliability and accuracy.

Among these autonomous functions, the core functions are related to decision-making and predictions and preservation. A decision-making-related function consists of aspects such as self-optimization, self-organization, self-decision-making, etc. Moreover, prediction- and preservation-related functions consist of aspects such as real-time monitoring, predictive maintenance, etc. Autonomous functions are operated based on certain factors, including the distributed control system, multi-agent system, AI, knowledge-base, big data, etc. SMSs exhibit similar properties because they inherited the structures and characteristics of NGMSs. The holonic manufacturing system (HMS), the biological manufacturing system (BMS), the intelligent manufacturing system (IMS), and the fractal manufacturing system (FrMS) are the most popular representatives of the next generation manufacturing system (NGMSs). Nevertheless, studies have focused on evolving IMS, HMS, and FrMS into SMSs via new technologies such as AI, big data, sensor data, etc. Among NGMSs, FrMS reported by Ryu et al. [12–14] exhibits suitable characteristics such as a distributed control system and a multi-agent system, using AI and the knowledge-base, is required for SMSs. Furthermore, FrMS features vertical and horizontal structures based on the fractal structure. It is based on the concept of autonomously cooperating multi-agents termed as fractals, which are referred to as the basic fractal unit (BFU) [12–16]. Specifically, BFU consists of an observer, a reporter, a resolver, an analyzer, and an organizer. BFU determines the goal decision, process optimization, and reconfiguration within the system's final goal. To determine the system's goal, goal-orientation technology is adopted through the goal decision model. The structure and process of the manufacturing system are subsequently altered by DRP. Self-optimization is conducted via BFU based on the knowledge base. Ryu et al. [12–14] were the first to propose the concept of a goal model for FrMS. However, FrMS needs to be improved and evolved into SMSs, including sustainable development. FrMS has sufficient potential and the fundamental background required to evolve into SMSs. However, thus far, approaches based on big data or active data utilization have not been adopted in FrMS in order to reconfigure the processes and the system.

Therefore, in this paper, we proposed an SSrMS architecture based on FrMS. SSrMS exhibits the features of a data-driven system. All the fractal units are operated based on the data in SSrMS. An external change is detected based on the environment information, which can serve as a signal to initiate a reconfiguration of the processes, structure, or the system itself. Internal changes are detected by the fractal units in SSrMS. This helps initiate negotiations or the re-optimizing process in the system. Furthermore, all the fractals exchange information, such as the current status, request, and response, to improve homeostasis and impart flexibility to account for changes. The optimization or reconfiguration results are stored in the knowledge base as a future reference for the next adaptation. For the data-driven system, SSrMS employs big data, which comprises the environment information, knowledge base, legacy system, and equipment signal DB. To determine the system's goal, each fractal's goal needs to be identified; thereafter, the process is reconfigured accordingly, and sustainability is assessed.

First, in order to decide the goal for the fractal and system, the goal decision model was developed. The goal decision model provides various types of goals for each fractal

and system, including productivity, cost, profit, etc. A neural network was used to develop the goal decision model, wherein it derives the goals of the system and each fractal. Thus, a goal-oriented system can be realized using the goal decision model.

Second, in order to reconfigure a manufacturing process, the negotiation model is developed. The negotiation model performs partial process optimization by reassigning tasks and resources between the fractals, based on the goal of coping with the changes in the system's condition.

Third, in order to create balance between the goal of the system and sustainability, the sustainability assessment method was developed. The sustainability assessment method was designed to simultaneously evaluate sustainability with respect to the system's goals using the balanced scorecard (BSC), the analytic hierarchy process (AHP), the green-bill of material, and neural network.

Given that the system and three methods are operated based on a data-driven system and big data, the proposed architecture and methods can serve as a fundamental basis for a new type of SMSs, one that is capable of self-optimizing and self-reconfiguring a system structure with manufacturing processes.

This paper aims to propose a new architecture and concept of smart manufacturing systems based on the fractal concept to achieve fully integrated autonomous operations. The proposed architecture supports providing a basic research foundation for self-reconfiguration, goal-orientation, and self-organization. The proposed models for goal decision and negotiation can enhance the possibility of facilitating goal-oriented and self-reconfigurable manufacturing systems. The sustainability assessment method proposed in the study can provide a new research direction to harmonize sustainability and the system's goal for the homeostasis of the manufacturing system.

This paper is organized as follows: Section 2 presents a literature review of existing manufacturing systems, such as SMSs, HMS, and FrMS, which have features of self-reconfiguration, self-optimization, and self-adaptation. In Section 3, the characteristics of SSrMS are presented. Section 4 presents the architecture of SSrMS with a sequence diagram of the manufacturing process reconfiguration. In Section 5, descriptions on the models for goal decision and negotiation as well as a sustainability assessment method are presented. Section 6 concludes this paper including further research.

## 2. Literature Review

### 2.1. Smart Manufacturing System (SMS)

In the 1980s, the basis of smart manufacturing was flexibility, computer-integration, and intelligent manufacturing. Japan established the intelligent manufacturing system (IMS) program in 1995 and has since led the world in this domain. In the US, non-profit ventures have conducted research on IMSs and related activities under the NGMSs program [3]. In the EU, studies on intelligent manufacturing were conducted under IMS program expansion [3,17]. NIST defines SMSs as a fully integrated collaborative manufacturing system that responds in real-time to satisfy the changing demands and conditions in factories, supply networks, and customer needs [2,3,18]. Different definitions exist for SMSs depending on the perspective: engineering, interconnection and communication, and predictive analysis and decision-making perspectives; this is illustrated in Table 1.

**Table 1.** Other definitions of the smart manufacturing system (modified from [2]).

| Perspective of Definition | Description |
|---|---|
| Engineering view | Enable rapid manufacturing of products, dynamic response to demands, and real-time optimization [19]. |
| Interconnection and communication (IoT and CPS) view | Increase in production rate and decrease in production waste using sensors and communication technologies to capture data at all manufacturing stages [20]. |
| Predictive analysis and decision-making view | Optimize planning and control of manufacturing operations, predictive manufacturing, fault diagnosis, etc. [21]. |

Various studies have focused on the SMS architecture; representative architectures for SMSs include the Industry 4.0 architecture by IBM, service-oriented smart manufacturing system by NIST, and RAMI 4.0 by Platform Industrie 4.0, as listed in Table 2 [22]. Industry 4.0 by IBM includes two layers: a platform/hybrid cloud layer and an equipment/device layer. The platform/hybrid cloud layer performs plant-wide data-processing and analytics, whereas the equipment/device layer acts as a middleman between the smart devices/tools and plant/enterprise. The service-oriented SMS by NIST is similar to Industry 4.0 by IBM [23]. The service-oriented SMS utilizes a service bus to connect various types of services, including those pertaining to the operational technology domain, information technology, and virtual domain. Furthermore, the service bus connects the enterprise to external collaborators. RAMI 4.0 consists of six layers: the business, functional, communication, integration, and asset layers. Each layer performs functions that define rules, integrate services, enable preprocessing with data analysis, provide event-generation with connectivity, and improve the physical performance of each component [24].

**Table 2.** Three representative architectures for SMSs.

| Architecture | Description |
|---|---|
| Industry 4.0 by IBM | Reference architecture comprising platform/hybrid cloud layer and equipment/device layer |
| Service-oriented SMS by NIST | Functional architecture utilizing a service bus to connect various types of services in the system |
| RAMI 4.0 by Platform Industrie 4.0 | Reference architecture comprising six layers with different functions, perspectives, and levels of controls. |

### 2.2. Holonic Manufacturing System (HMS)

Koestler defined a holarchy as a hierarchy of self-regulating holons in supra-ordination to their parts, sub-ordination to the higher levels, and coordination with an environment [25]. The HMS consortium converted the concepts developed by Koestler for social organizations and living organisms into a set of appropriate concepts for manufacturing industries [26]. Over the last two decades, many research results and developments have been reported for HMS. Additionally, following the paradigm shift caused by Industry 4.0, HMS was also attempting an effort toward smartization. For the evolution to Industry 4.0, ten key enablers were selected for the HMS: sustainability, secure communication/cyber-resilience, real-time capabilities, process virtualization, service orientation, interoperability/integration, adaptability, big data analysis, autonomous and decentralized decision support systems, and connectivity [27]. With these ten key enablers, Leitao and Restivo proposed the adaptive holonic control architecture (ADACOR), which features a decentralized control architecture and considers centralization to tend toward a global optimization system [27]. Pach et al. [28] proposed the optimized and reactive control dynamic architecture (ORCA), which is one of the first dynamic architectures. Furthermore, Barbosa et al. [29] proposed the evolution of the ADACOR mechanism into ADACOR$^2$. Specifically, the objective of ADACOR$^2$ aims at allowing the system to evolve dynamically via configurations discovered online, which are limited to the stationary and transient states alone.

### 2.3. Fractal Manufacturing System (FrMS)

As mentioned before, FrMS is based on the concept of autonomously cooperating multi-agents, referred to as fractals. FrMS exhibits several characteristics, including self-similarity, self-organization, and goal-orientation, and so on [15,16]. Furthermore, the definitions of a fractal and FrMS are as follows.

- A fractal is a set of self-similar agents whose goal can be realized through cooperation, coordination, and negotiation with others. It can reorganize the configuration of the fractal system to realize a more efficient and effective configuration;

- FrMS is a flexible and fault-tolerant system developed and operated with a fractal architecture.

The FrMS is an agent-based system that continuously reorganizes the system configuration to remain in an optimal environment. The agent-based system can autonomously improve its structure, such as upgrading servers, moving services, and performing load balancing interposed without interruptions or revisions to the network and clients. Figure 1 illustrates the concept of FrMS. The architecture of FrMS comprises a hierarchical structure developed using the elements of BFU, which is a fundamental component of the FrMS. As shown in Figure 2, The five functional modules in the BFU are the analyzer, organizer, observer, resolver, and reporter. The observer module gathers information or messages from other fractal agents and the environment. Subsequently, it delivers this information to the analyzer module and resolver module. The analyzer module obtains this information from the observer module and analyzes information, such as scheduling, simulation, and changes in the fractal's objective. The resolver module then makes alternative decisions for current fractals, such as new objectives or negotiations with another fractal, based on the analyzed information and message derived from the analyzer and observer modules. The organizer module checks the current status of the fractal, and it can alter the fractal structure. The organizer module then transmits the data related to the status and altered structure to the analyzer and resolver modules. Lastly, the reporter module reports all types of data and information to the other fractals or sensors.

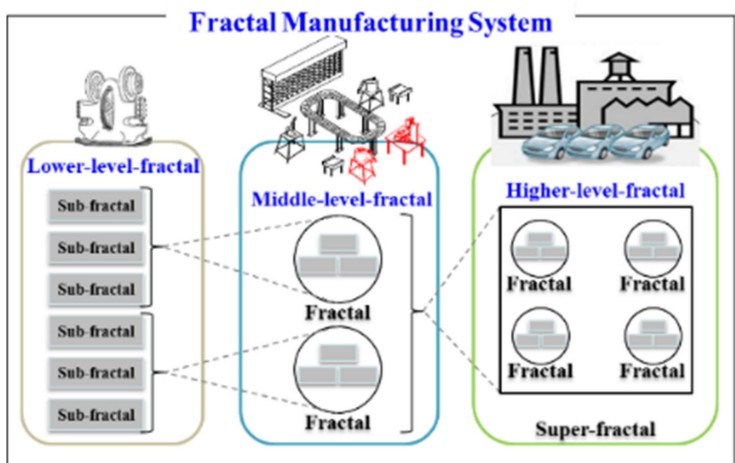

**Figure 1.** Conceptual model of FrMS.

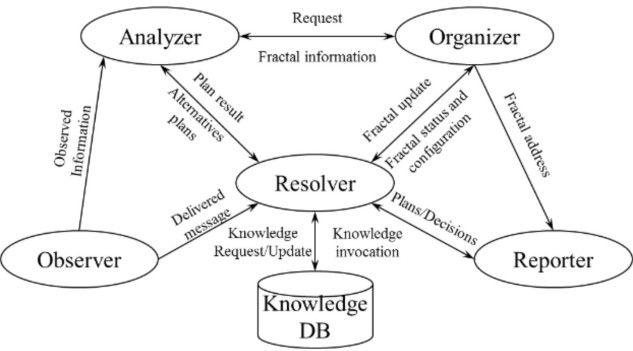

**Figure 2.** Five functional modules in a BFU.

### 2.4. Goals in Manufacturing Systems

Studies on the objectives, goals, and evaluation methods for manufacturing systems have been popular since the establishment of the manufacturing industry. Studies on manufacturing systems' objectives can be classified into three categories: key performance index (KPI), objectives definition, and new evaluation method. However, in all existing studies,

the objectives and goals of the manufacturing system have always been considered as essential and key factors. Studies on the objectives and goals have afforded many types of KPIs for evaluation and optimization purposes. Hon [30] stated that the evaluation factors for manufacturing systems exhibit different trends with respect to time and issues. In the 1960s, the priority of manufacturing systems was cost minimization. By contrast, in the 1970s, productivity was the most important KPI. Furthermore, once the mass production system was established, in the 1980s quality was considered as the more important KPI, rather than productivity. In the 1990s, the optimization of manufacturing systems was studied based on a multi-dimensional perspective. Currently, sustainability, eco-friendliness, and zero-emissions are considered as the most important KPIs. Craig and Harris [31] and Son and Park [32] stated that cost is the primary priority for manufacturing systems. Hopp and Spearman [33] hierarchically categorized the goals of manufacturing systems. Furthermore, Hopp and Spearman suggested profit maximization as the priority. Troxler [34] defined four attributes—suitability, capability, performance, and productivity—consisting of 24 KPIs to evaluate a system based on the traditional perspective. Since the 1990s, either profit or performance has been selected as a manufacturing systems' objective and goal. Jose et al. [35] listed the top five parameters in a manufacturing system as profitability, conformance to specifications, customer satisfaction, return on investment, and material and overhead costs. Furthermore, the least significant parameters were efficiency, quality, technical competence, flexibility, innovation, speed, and capacity. In the 2000s, new objectives and goals were examined, including business strategy, innovation, sustainability, eco-friendliness, etc. Golec and Taskin [36] reported that a manufacturing system's goal consists of nine parameters: innovation, customization, product proliferation, price reduction, cost, flexibility, quality, speed, and stability. The objectives and goals of a manufacturing system were defined by Avella et al. [37]; they analyzed the relationship among parameters in order to measure the capacity of the manufacturing system.

### 2.5. Negotiation Methods for Manufacturing

Krothapalli and Deshmukh [38] proposed a negotiation methodology that negotiates between parts and machine using internal and external agents in the manufacturing system; this methodology enables easier negotiations. However, Krothapalli and Deshmukh did not consider the negotiation between multi-agents. Shin and Jung [39] proposed a mobile agent-based negotiation process (MANPro), which offers a negotiation process for a manufacturing system with intelligent distributed control. MANPro consists of part-oriented, machine-oriented, and bi-directional bidding for improved negotiation performance. Furthermore, MANPro involves a mobile agent technique, wherein a mobile agent can travel between systems. Shin and Jung [40] proposed a methodology for negotiation process creation, evaluation, and real-time scheduling based on MANPro; however, they also failed to consider the negotiation between multi-agents. Adhau et al. [41] proposed the auction-based negotiation methodology, in which multiple projects can be scheduled simultaneously. This methodology can solve scheduling optimization for multiple projects under resource limitations. Furthermore, many agents can participate in the negotiation simultaneously under the auction method. However, one agent utilizes the resources first, and the other agents subsequently follow the auction procedure again; moreover, the resources for the winner agent cannot be re-allocated to other agents. Gordillo and Giret [42] proposed a multi-agent scheduling algorithm to evaluate manufacturing systems based on the negotiation methodology. This algorithm considers job priority changes and job allocation under various situations. Furthermore, it can perform rescheduling at any instance; however, it does not consider resource utilization. A comparison of previous research shows that process reconfiguration has not been considered in existing studies. Krothapalli and Deshmukh considered the negotiation between single agents. Shin and Jung considered the negotiation between multi-agents with single-agents, and they also included a scheduling algorithm that could re-allocate the task sequence. Therefore, the methodology proposed by Shin and Jung covered the aspect of process reconfiguration.

Adhau et al. [41] considered the negotiation between multi- and single-agents. However, the methodology of Adhau et al. [41], i.e., the resource allocation method, is not suitable for process reconfiguration. Gordillo and Giret [42] considered the negotiation between multi-agents and that between multi- and single-agents.

### 2.6. Sustainability for Manufacturing

Sustainability for manufacturing is implemented by process efficiency, increasing resource efficiency, reducing power consumption, increasing automation for human labor, etc. [43,44]. In order to improve the sustainability in the manufacturing system, eleven critical success factors (CSFs) that satisfy the requirements of both SMs and environmental sustainability are suggested by Jabbour et al. [45]. Existing studies on sustainability and its assessment use the analytic hierarchy process (AHP) and outranking. AHP and outranking methods determine an optimal alternative considering a given circumstance. In order to achieve project management with sustainability, a functional redundancy-based human resource management framework is suggested by Dotsenko et al. [46].

Another method to assess sustainability is the heuristic approach based on the sophisticated mathematical model [47].

However, a single objective or restricted goal is considered by existing sustainability assessment. Furthermore, existing sustainability assessments do not consider sustainability and system's goal at the same time.

## 3. Characteristics of Smart Self-Reconfigurable Manufacturing System (SSrMS)

### 3.1. Self-Similarity

Self-similarity is a unique characteristic of the fractal structure, where the entire shape and partial shape are similar. In FrMS, self-similarity is adopted for the structure and organization design and job, formulation, and goals [12,14]. Figure 3 shows the self-similarity structure with the facility for SSrMS. The super-fractal in Figure 3 is the highest fractal unit in the system. The other fractals belong to the super-fractal. Logically, each fractal has the same structure. Figure 3 presents the hierarchical structure. Self-similarity can afford simplified logical structures and ease of building. Additionally, the same logical fractal structure can easily add, remove, or replace the equipment in the system.

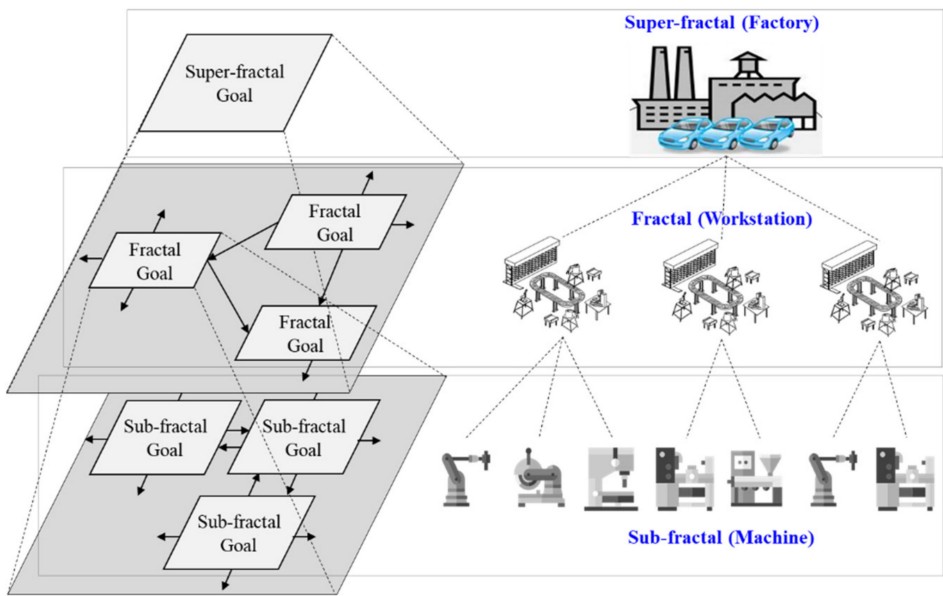

**Figure 3.** Self-similarity and hierarchical fractal structure.

### 3.2. Self-Organization

Self-organization refers to the ability whereby each fractal unit can develop the manufacturing process by itself via goal-orientation. Each fractal has sufficient information

regarding each fractal's performance data, the precedence relation, and the system's goal. Thus, the system can organize the manufacturing processes after deciding the goal, as illustrated in Figure 4. Furthermore, self-organization can help reorganize the manufacturing process when the system's goal changes or the system is optimized. Moreover, DRP can address the reorganized manufacturing process, as illustrated in Figure 4.

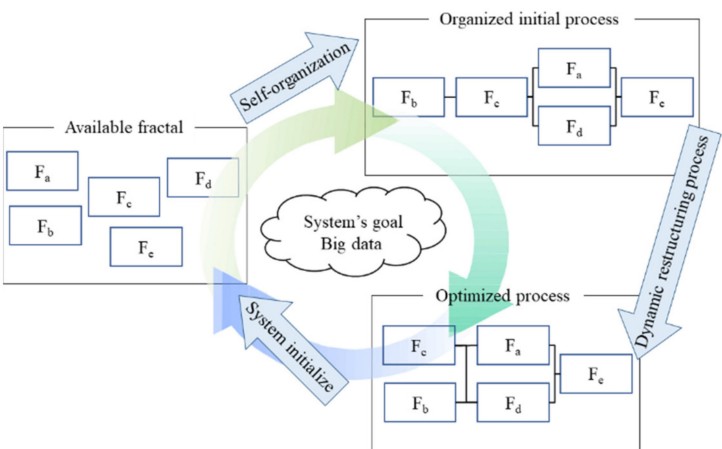

**Figure 4.** Self-organization process and DRP in SSrMS.

### 3.3. Goal-Orientation

SSrMS consists of fractal units with self-similarity. If the highest-level fractal can decide the goal by itself, then the other fractals can also determine the goal by themselves, because every fractal has the same logical structure. Hence, the goal-orientation characteristic enables each fractal to determine an individual goal within the system's goal. Notably, the system's goal needs to be changed based on the circumstances. Furthermore, the system needs to adjust the goal when the fractal exhibits a state change, as follows: equipment malfunction and addition or removal of the fractal. Ryu et al. [13,14,48] suggested a goal-formation process for the goal-orientation characteristic, as illustrated in Figure 5.

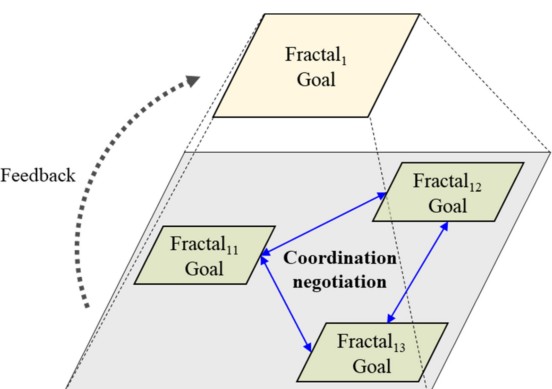

**Figure 5.** The goal-formation process by Ryu et al. (modified from [13,14]).

### 3.4. Self-Learning

To perform self-reconfiguration in the system, the system needs to conduct better decision-making or at the very least recommend a better alternative. This type of advanced methodology can control the structure and also monitor and modify the structure of the manufacturing system. Over the past several decades, self-learning, self-optimization, and self-evolution have become important and active research areas. Among these, self-learning is one of the critical characteristics for the reconfiguration of a manufacturing system. The system requires self-learning to improve its performance. Each fractal can decide on the goals, including appropriate manufacturing processes between the fractals, operation performance within the highest goals, and self-diagnosis. Therefore, in SSrMS,

self-learning implies that each fractal can save and learn from past decisions and thus perform better decision-making, as compared to previous decisions. The previous decisions correspond to the goal with the fractal status, performance with the manufacturing process, negotiation results with the performance, self-diagnosis with malfunctions, and fault results with a fractal condition, as illustrated in Figure 6. To perform self-learning in the system, the system requires big data or a data lake. As shown in Figure 6, the fractal gathers information from big data and the circumstances. The fractal then decides the required changes based on data, such as the manufacturing process, selection of new goals, and reassignment of resources between fractals. Subsequently, the fractal searches for a new result to ensure better performance as compared to the previous performance of the fractal itself. After applying the new result, the fractal analyzes and compares the new result with the old decision information from the knowledge base using AI, heuristics, and reinforcement learning. Following this feedback, the knowledge base now contains new decision information. If the fractal requires a new decision to ensure better performance, the fractal employs the old decision information from the knowledge base. Furthermore, the fractal will repeat these steps with a learning algorithm. The entire fractal can share and use the information from the knowledge base.

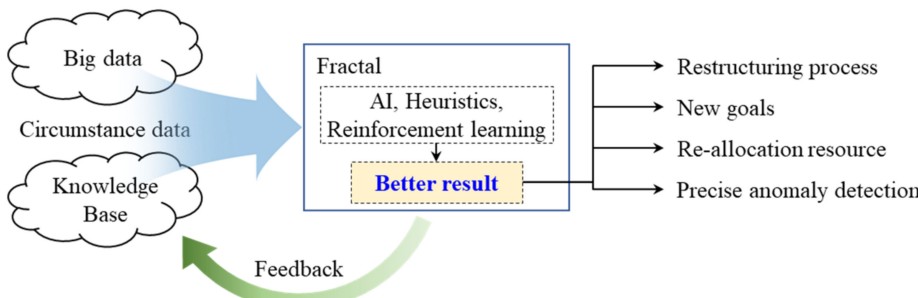

**Figure 6.** Self-learnable characteristic for SSrMS.

### 3.5. Self-Decision-Making

Self-decision-making refers to the ability whereby each fractal can decide the behavior, goals, and negotiation between fractals. For this purpose, a fractal first receives a signal that represents a request for adopting a new goal, restructuring processes, or reassigning resources. Thereafter, the fractal can execute the appropriate job, such as a derived result. Based on the derived result, the fractal decides the following behavior. The self-decision-making characteristic is a fundamental feature of SSrMS because most of the technologies in the SSrMS require the self-decision characteristic. However, self-decision-making does not imply a fully autonomous decision-making technique. To prevent unexpected errors, self-decision-making needs to be executed within a limited range, such as by lowering the fractal's goal decision, negotiating between fractals, adding/removing fractals based on equipment failure, etc.

### 3.6. Self-Regulation

Self-regulation is a characteristic of a data-driven system, and its role involves controlling the equipment based on internal data. In SSrMS, each fractal checks the status data from equipment in real time to adopt appropriate regulations, as illustrated in Figure 7. For example, when an unusual operation is detected by a fractal from the signal data based on prediction, the fractals can request a job change or cease operation for maintenance of equipment. Additionally, the fractals can appropriately control equipment for resource utilization based on the energy consumption data, Green-BOM data, and inventory data, etc. Self-regulation is performed via a fractal, which represents the equipment. If a fractal is removed or unavailable, self-regulation can lead to goal changes or process reconfiguration in the system. This implies that the regulation triggering conditions in the system are the abnormal signals detected from equipment due to factors such as equipment malfunction,

detection of product defects, addition/removal of adjacent equipment, reconfiguration messages, and demands to alter the goal.

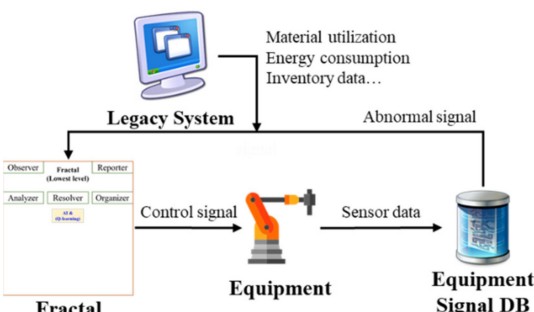

**Figure 7.** Self-regulation characteristic for SSrMS.

### 3.7. Self-Execution

Self-execution is aimed at controlling equipment via internal and external data. For this, the fractal checks the consumption data of adjacent equipment based on the current equipment and remaining resource level data. Furthermore, environmental information is used to predict market trends and material prices based on analyses of web data, as illustrated in Figure 8. Based on the prediction results, the analyzer in the fractal compares the prediction result and the current status. If required, adjustments in the facility and scheduling are requested by the resolver in the fractal. Thereafter, the higher-level fractal receives this adjustment request from the lower-level fractal, which is responsible for the equipment. The higher-level fractal then decides on process reconfiguration via the negotiation model. Therefore, self-execution is conducted by the higher-level fractal. Additionally, using self-execution, the higher-level fractal can request a change in the goal via forecasting based on certain factors, including market trends, stock market trends, and customer demands.

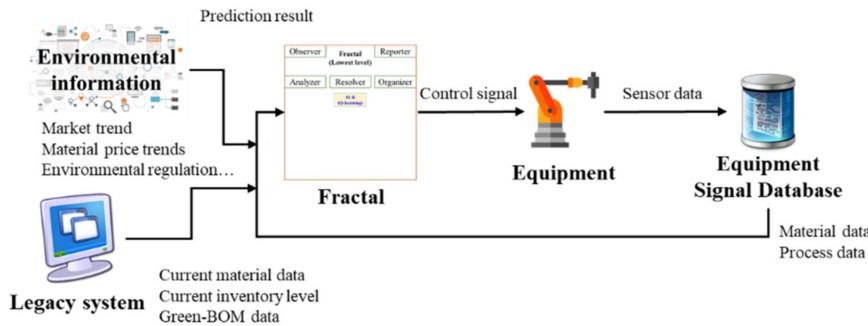

**Figure 8.** Self-execution characteristic for SSrMS.

## 4. The Architecture of the Smart Self-Reconfigurable Manufacturing System (SSrMS)

### 4.1. SSrMS Architecture

Currently, the importance of acquiring and utilizing appropriate data for the optimization and development of manufacturing systems is increasing. Most analyses, optimizations, and autonomous functions for SMSs are based on data that can be obtained from various sources, including the equipment, legacy system, environment, etc. Therefore, SSrMS also involves the concept of data-driven manufacturing systems.

Based on the characteristics of SSrMS, Figure 9 shows the proposed architecture of SSrMS. The proposed architecture consists of big data, big data analysis tools, digital twins (simulation), and a fractal structure. The fractal structure exhibits a recursive shape, where the highest-level fractal (super-fractal) consists of two fractals (sub-fractals). Furthermore, these fractals comprise the lower-level fractal (lowest-level). Each fractal has five agents: the observer, reporter, analyzer, resolver, and organizer. Additionally, SSrMS incorporates an AI and Q-learning module to support the resolver agent. It executes the search for

new goals, negotiation between fractals, optimization of fractal status, self-learning, and restructuring processes. Digital facilities and simulations are some forms of the digital twin. Before SSrMS adopts a new goal, structure, or manufacturing process, digital facilities and simulations are used to simulate the digital facilities with the derived solution. These digital facilities can help improve the stability of the factory, reduce accidental losses, and improve the self-learning accuracy.

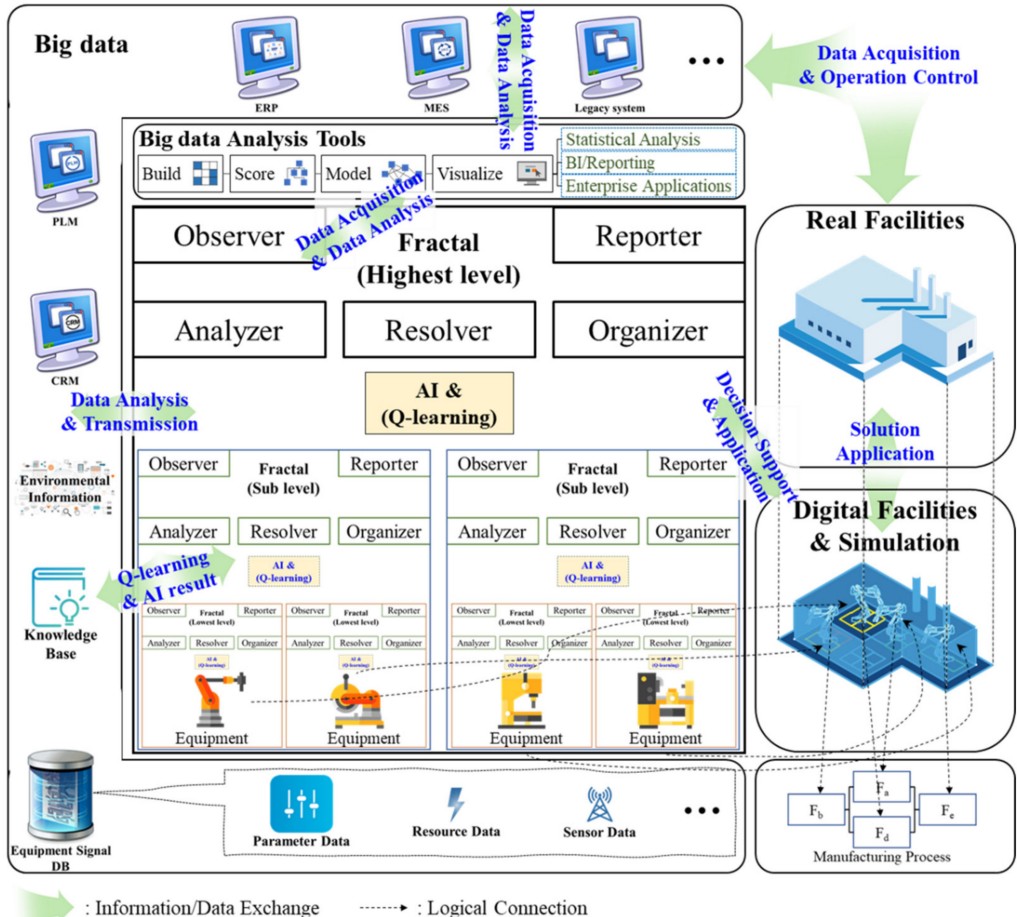

**Figure 9.** Architecture of SSrMS.

Big data comprise equipment data, the knowledge base, environmental information, and the legacy system data. Equipment data are collected using the equipment signal of the database (DB), which consists of the sensor data, parameter data, power utilization data, resource utilization data, etc. The knowledge base includes previous reconfiguration result data, Q-learning result data, negotiation result data, metadata of the trained AI, each fractal's optimization status data, etc. The knowledge base contains essential information and data required for the smartization of the self-reconfiguration manufacturing system. Furthermore, environmental information includes significant amounts of ambiguous information from the real-world, such as material price trends, stock market information, new technology news, and other information that can affect the manufacturing system. Thus, it is difficult to clean this environmental information and transform it into information. Hence, data collection technologies are required to acquire data from the Internet, and web-based survey techniques using AI are adopted to obtain useful environmental information. This approach can help predict the condition of the system in the future. Based on the prediction, suitable pathways to realize the system's goal can be suggested. In this manner, environmental information affords better reconfiguration directions for the manufacturing system using the other data included in big data.

Figure 10 illustrates the data-driven modeling of big data for SSrMS. The data required for process reconfiguration or goal changes are provided by the legacy systems, environment information, and equipment signal DB. Process reconfiguration and goal change are conducted by the goal decision model, negotiation model, and sustainability assessment method in the fractal. The result of this process reconfiguration and goal change is then restored in the knowledge base. Digital facilities execute simulations to verify the results from fractals. Furthermore, actual facilities apply the process reconfiguration and goal change using fractals. These new goals and manufacturing processes are then updated at all the fractals.

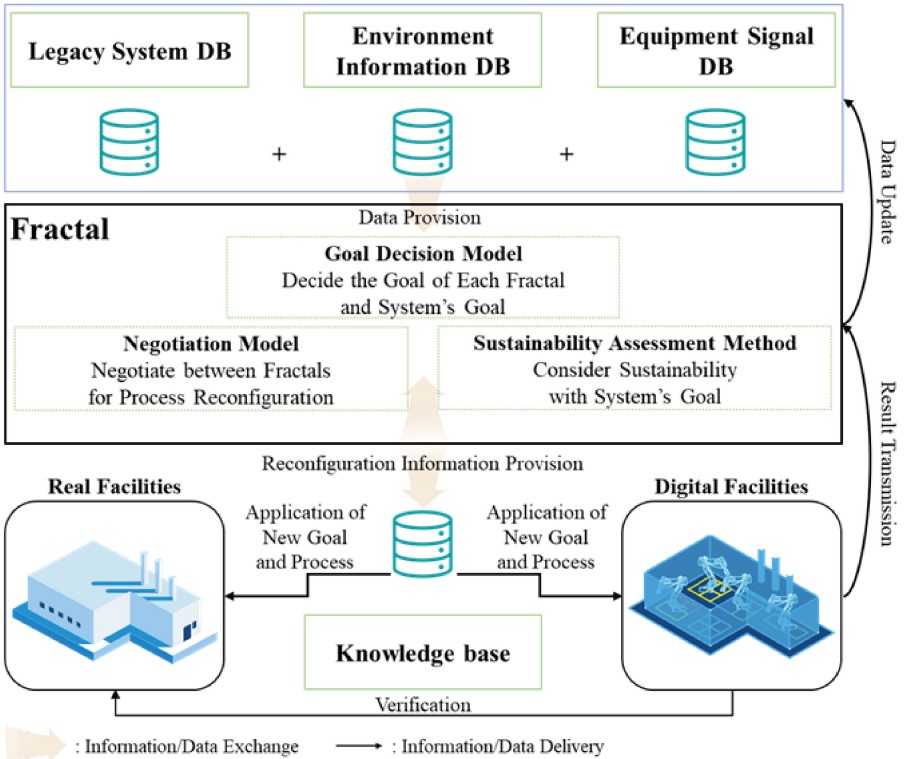

**Figure 10.** Data-driven modeling of big data for SSrMS.

The method of acquiring data and synthesizing relevant information from the external data is based on the environment information, as illustrated in Figure 11. The collection of web data is performed using AI, which can determine valuable data, including market trends and material price trends. A prediction method based on AI assesses the future trends of the market, material prices, etc. Furthermore, the highest-level fractal, as a representative of the factory, receives the prediction results and performs the analyses.

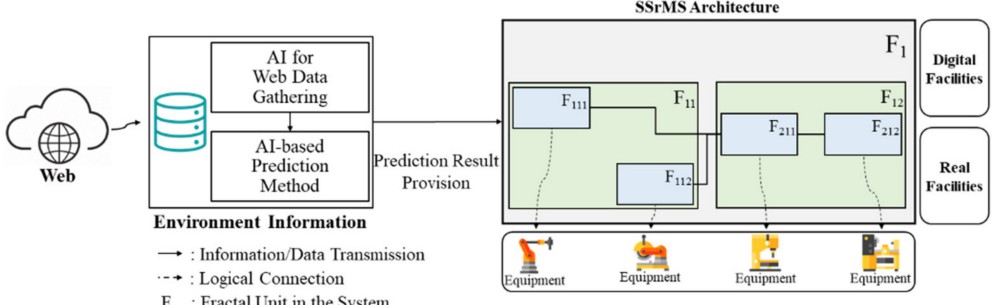

**Figure 11.** Environment information in SSrMS.

The legacy system data comprise the data from sources such as product lifecycle management (PLM), enterprise resources planning (ERP), customer relationship manage-

ment (CRM), and the manufacturing execution system (MES). These legacy system data can provide important information, including operation scheduling, planning and actual operation data, the system's current status, overall equipment effectiveness (OEE), resource information, etc. With legacy system data, the manufacturing system can compare the planning and actual situations of the system and search for the reconfigurable spot to realize the goal.

Big data analysis tools handle the equipment signal DB, environmental information, and legacy system data. On receiving an information request from a fractal, the big data analysis tools collect data from the environment and analyze the data based on the request. Specifically, large amounts and different types of signal data and legacy system data exist. Therefore, the system requires an application to efficiently handle this vast amount of data.

### 4.2. Sequence Diagram of the SSrMS Architecture

The self-reconfiguration procedure in SSrMS architecture is realized by BFU, as shown in Figure 12. The definition of BFU is similar to that for FrMS. The self-reconfiguration procedure commences from the big data or operator's request. SSrMS recognizes the reconfiguration request information. The observer in BFU then receives this request and sends it to the analyzer. The analyzer decides to either receive or refuse the request; the analyzer refuses the request if the fractal is not subject to reconfiguration, malfunctions, or insufficient capacity. On accepting the request, the resolver receives the information to be analyzed and decides on the type of reconfiguration suitable for the fractal. Hence, the resolver decides the reconfiguration method based on the restructuring structure and sets up the new goal for SSrMS and the reconfiguration processes. Furthermore, the resolver determines the other participants (usually lower-level fractals) required for the reconfiguration. Once the participants are decided, the reporter receives a message from the resolver, which requests other fractals to participate. The reporter then calls the observer of the other fractals to request joining reconfiguration. Subsequently, the resolver calls the organizer to check the status of all fractals. The organizer sends a message to the resolver confirming the status of all fractals. Thereafter, the resolver requests the AI module for the execution of an optimization job. The AI module requests the previous optimization results and data from the knowledge base. The knowledge base then sends the requested data to the AI module and resolver. Based on these data from the knowledge base and the data from the data lake, the AI module begins to build the model, performs training, and obtains the result.

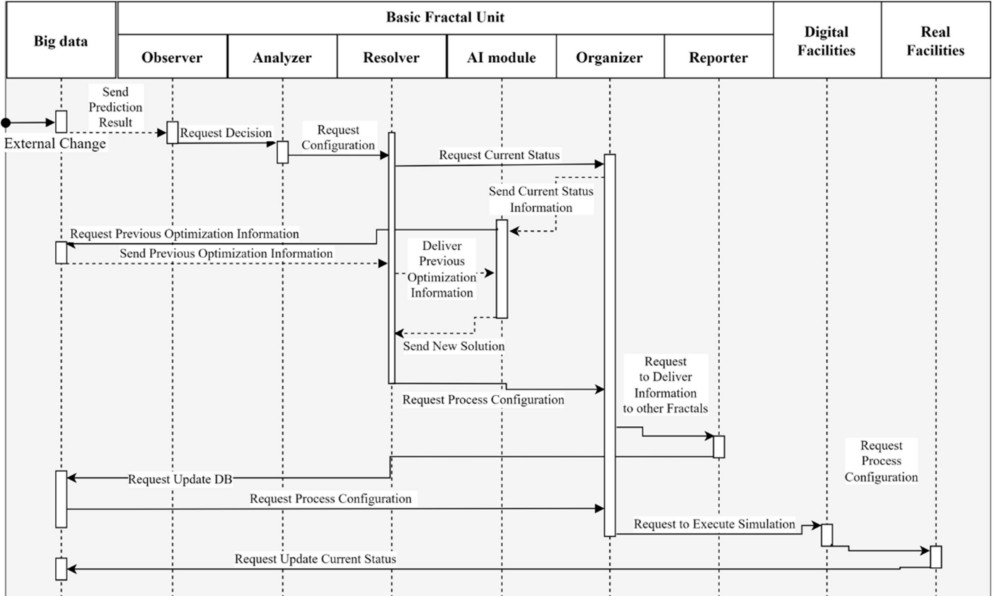

**Figure 12.** Sequence diagram of reconfiguration flow in SSrMS.

Meanwhile, the resolver attempts to obtain the optimization solution using heuristics and algorithms. Thus, all the work is performed by the resolver and AI module, wherein the resolver receives all the information pertaining to the results and decides on the optimal solution. The optimal solution thus obtained is stored in the knowledge base. Furthermore, the resolver temporarily calls the organizer to apply the new reconfiguration method. The organizer sends the new reconfiguration solution directly to the digital facilities. Thereafter, the digital facilities perform simulations of the digital factory using the new solution. Based on the simulation results, the resolver assesses the performance of the system. Finally, if the necessary conditions are satisfied, the resolver makes the final decision to apply the new reconfiguration solution to the actual factory.

## 5. Self-Reconfiguration Methods in SSrMS

### 5.1. Goal Decision Model

There are three methods for process reconfiguration which are the goal decision model, negotiation model, and sustainability assessment method in SSrMS.

Ryu and Jung [12] suggested the concept of a goal decision model to generate a system's goal based on environmental needs. The goal decision model falls under the category of goal-orientation technologies, which consist of GGP, GHP, and GBP. However, the reference goal decision model cannot represent all the goals in-depth because the goals for the manufacturing system are complicated owing to the interdependencies among the goals. Lee et al. [49,50] suggested a goal decision model mechanism based on the goal model concept of Ryu and Jung [12]. The goal decision model is a crucial mechanism for determining the system's goal. Furthermore, the proposed model features flexibility in terms of deciding goals. The mechanism of the goal decision model includes five steps, as illustrated in Figure 13. As shown in Figure 13, the goal decision model can determine the system's initial best goal.

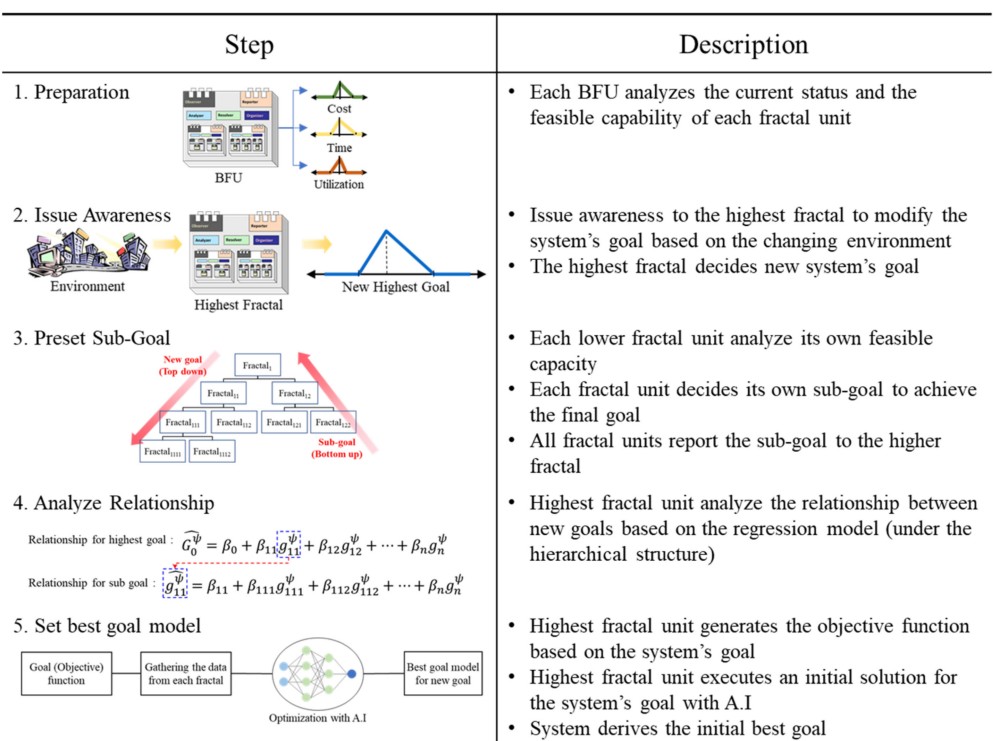

**Figure 13.** Mechanism of goal decision model [49].

In order to perform experiments, a neural network (NN) model has been developed for implementing the goal decision model by using Python 3.8. To develop the NN model, the results of NN optimization provided by the Matlab library have been considered. As shown

in Figure 14, the activation function of the NN is a Gaussian function by data tendency verification from Matlab. Therefore, the NN includes a regression model with a Gaussian activation function. The total number of data was 500 and sizes of data for testing and validation were 50% each. The number of training epochs was 1000. Data used in this study are artificially made, similar to actual company data. This is because actual data have many missing values and it may have a negative impact on training and validation. The NN consists of one input layer (5 nodes), four hidden layers (32 nodes-64 nodes-32 nodes-5 nodes), and one output layer (1 node). As shown in Figure 14, data for the experiment consists of productivity, efficiency, utilization, flexibility, cost, and profit. The values of productivity, efficiency, utilization, flexibility, and cost range from 0 to 1, but that of profit ranges from 0 (i.e., zero profit) to 10 (i.e., maximum profit). As illustrated in Figure 14, each fractal has its own NN to realize its goal, and the result from each fractal can be used as an input value to realize the system's goal. Based on the derived value of the system' goal, the NN can obtain each fractal's optimized parameter value. Based on the NN structure, individual fractals can decide a new goal under the system's goal in order to reset the goal. Furthermore, the manufacturing system can reset the goal or execute partial reconfiguration in the system using the proposed model. However, the signal and process data cannot be easily obtained for experiments. Therefore, the experiment is limited within the dotted box illustrated in Figure 14.

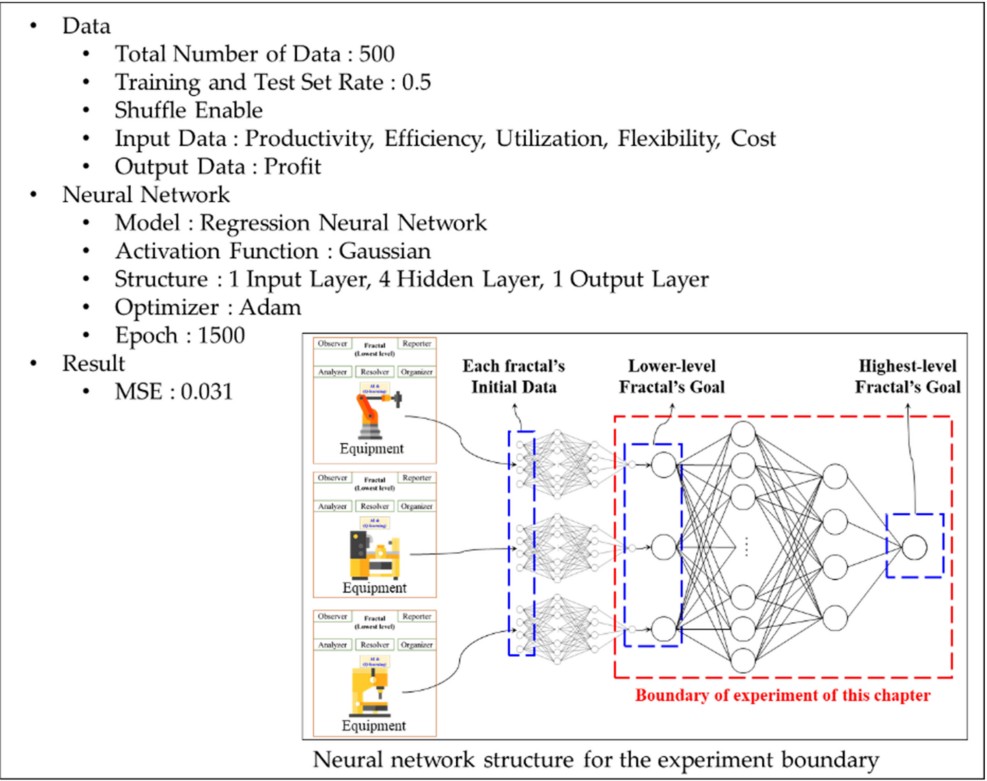

**Figure 14.** Overview of experiment for the goal decision model.

As mentioned before, the size of data for testing and validation were 50% each, and 1000 training epochs were adopted. After training the NN, the validation mean squared error (MSE) value for prediction was 0.031. Furthermore, the predicted initial solution value from the trained NN was 5.38. When ten is the maximum profit ratio without any constraints, the value of NN result means the best profit ratio under the constraints, such as cost, equipment capability, time, and so on. The basic experimental assumption was that the exact relationship and equation of the data were unknown. However, to verify the trained neural network, the original and experiment results were compared, as illustrated in Table 3. The normalization process can provide unified units for different units, such as percentages,

time, etc. The values in Table 3 are within the range of 0 to 1 or 0 to 10; these values were normalized from the original data. Therefore, a goal for the system can be derived after converting the normalized values to the actual values. The difference between the actual values and the neural network results is 5.28%. In this experiment, this result indicates that in the real world, manufacturing systems include structured and unstructured data, such as signal data from equipment, process data from the legacy system, image data for quality inspection, text data, multimedia data, etc. Furthermore, the number of data points is extremely high. Given these conditions, a neural network is suitable to derive the goals for actual manufacturing systems.

**Table 3.** Actual and experimental result.

|  | System's Goal (Profit Value) | Difference (%) |
|---|---|---|
| Actual value | 5.61 | |
| Neural network result | 5.38 | 5.28% |

The goal decision model in SSrMS functions as illustrated in Figure 15. Each piece of equipment in the factory is connected with each fractal. The highest-level fractal detects the need to change the goal of the system and first decides the system's goal. Then, the lower-level fractals decide their goals to realize the system's goal. A neural network is used in the goal decision model to determine each fractal's goal. Furthermore, the system applies the new goal to the entire system and each fractal based on the neural network results.

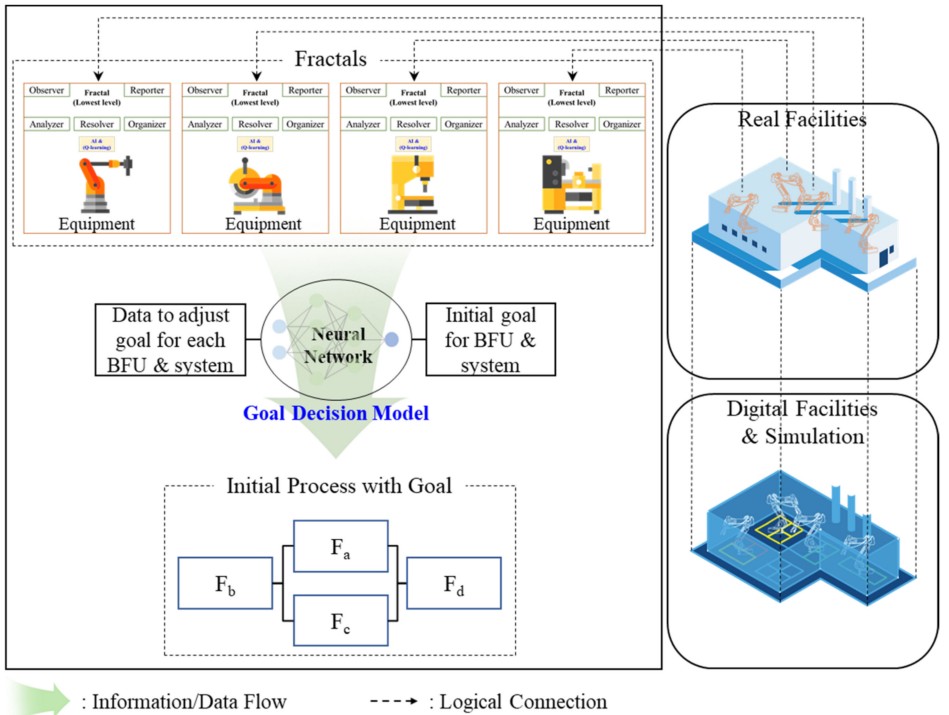

**Figure 15.** Goal decision model in SSrMS.

### 5.2. Negotiation Model

The negotiation model can reduce the time and cost of reconfiguration by reconfiguring a part of the manufacturing process. Furthermore, it is possible to realize the system's goal by considering each fractal's goal. Lee et al. [51] suggested a negotiation model for reconfiguring the manufacturing process with many negotiations, as illustrated in Figure 16. The method proposed by Lee et al. [51] consists of three mechanisms: resource-based negotiation, task-based negotiation, and hybrid-type negotiation. The hybrid-type negotiation involves the use of the genetic algorithm (GA) to realize the optimization

process. To execute hybrid-type negotiations, Lee et al. [51] suggested five agent modules: the bid-manager, agent dispatcher, task agent module, negotiation agent module, and resource agent module.

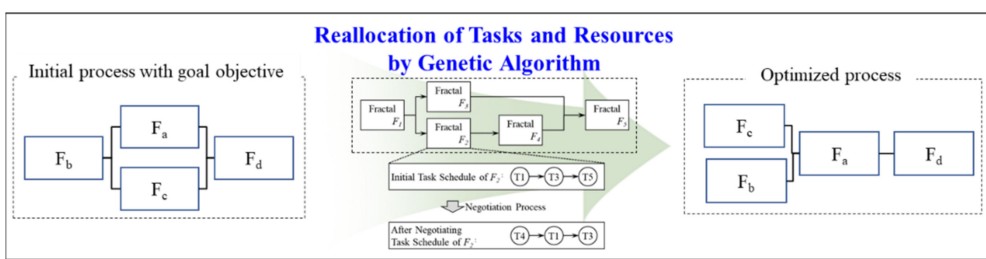

**Figure 16.** Negotiation model in SSrMS.

The hybrid-type negotiation can simultaneously reassign tasks and resources using the GA. In the hybrid-type negotiation method, the GA derives optimized reallocation. Based on the results of reallocation from the GA, hybrid-type negotiation realizes optimization. The hybrid-type negotiation involves three steps to perform negotiations with the GA as shown in Figures 17 and 18. Steps 1 and 2 of the hybrid-type negotiation process are shown in Figure 17. Step 1 begins with the bid-manager determining the fractal and then joining a negotiation process. The bid-manager decides the candidates for negotiation, and each candidate checks their sub-fractals. If the candidates have sub-fractals, then all the sub-fractals enter the negotiation process. All the fractals also check priority to confirm the negotiation participants once the bid-manager determines all candidates. Fractals with insufficient priority are excluded from the negotiation. Based on step 1, the fractals are determined to participate in the negotiation. In step 2, the final boundary is determined according to the highest-level fractal. The highest-level fractal determines the final boundary of the negotiation.

In step 3, the hybrid-type negotiation process is performed, as shown in Figure 18. Hybrid-type negotiation covers two cases: negotiation case 1 and negotiation case 2. Negotiation case 1 refers to the negotiation between multi- and single-agents. Furthermore, negotiation case 2 refers to the negotiation between multi-agents. A single fractal shares the resources or tasks under negotiation case 1. By contrast, under negotiation case 2, multiple fractals assign the resources and tasks using the GA.

In order to perform the experiment by GA, the GA was implemented using Java Eclipse to verify the hybrid-type negotiation. The population size was set as 100 units, and the number of iterations was 700. The mutation ratio was set as 0.1. In the GA, a chromosome consists of two parts indicating the fractal ID and the tasks with sequence. A gene in the chromosome has two types: fractals and tasks. A fractal gene has information on unique fractal IDs, which represent each piece of equipment including $F_1$, $F_2$, and $F_3$. A task gene has information on the job and task, which represents the job and task IDs (e.g., $J_1$, $J_2$, $J_3$, ... and $T_{11}$, $T_{12}$, $T_{21}$, $T_{31}$, ... ). Furthermore, each task has information regarding the processing time of every other task. The GA has three objectives, which are the minimum total task completion time for each fractal (i.e., *CF*), the minimum workload of each fractal (i.e., *WF*), and the minimum total workload of fractals (i.e., *WT*).

The fitness value of the chromosome is decided by the objective function $f(z)$, which corresponds to min $f(z) = (CF + WF + WT)$. Each objective in the fitness function is applied with identical weights. The fitness function is applied to evaluate the fitness of all the chromosomes in the population. For the fitness evaluation, *CF*, *WF*, and *WT* are computed for each chromosome in the current generation. If the new chromosome has a higher fitness value, then the new chromosome is chosen. The recombination method is a two-point crossover, and fitness selection is a roulette wheel as a random. Two-point crossover was applied for fractal selection. Two-point crossover covers a larger space than one-point crossover. Furthermore, two-point crossover is more suitable for solving this problem. The order crossover method was used for the task sequence. Additionally, the mutation of

the GA only altered the assignment property of the chromosomes for the fractal selection and task sequence. The experiment's objective corresponded to the minimization of workload. Data pertaining to the ten fractals and tasks were assumed to be available for the experiment. To observe the results more clearly, it was assumed that certain tasks impose heavy workloads on specific fractals as a penalty. For instance, $T_{23}$ requires 54 min in $F_3$ to constitute a disadvantage, as illustrated in Table 4. The five fractals join the negotiation process with four tasks. Hence, $F_1$ operates $T_{21}$ in $J_2$, $T_{41}$ in $J_4$, and $T_{23}$ in $J_2$, whereas $F_3$ operates $T_{31}$ in $J_3$. In this experiment, the workload reduced to 32 min, and the minimized total workload was 8 min.

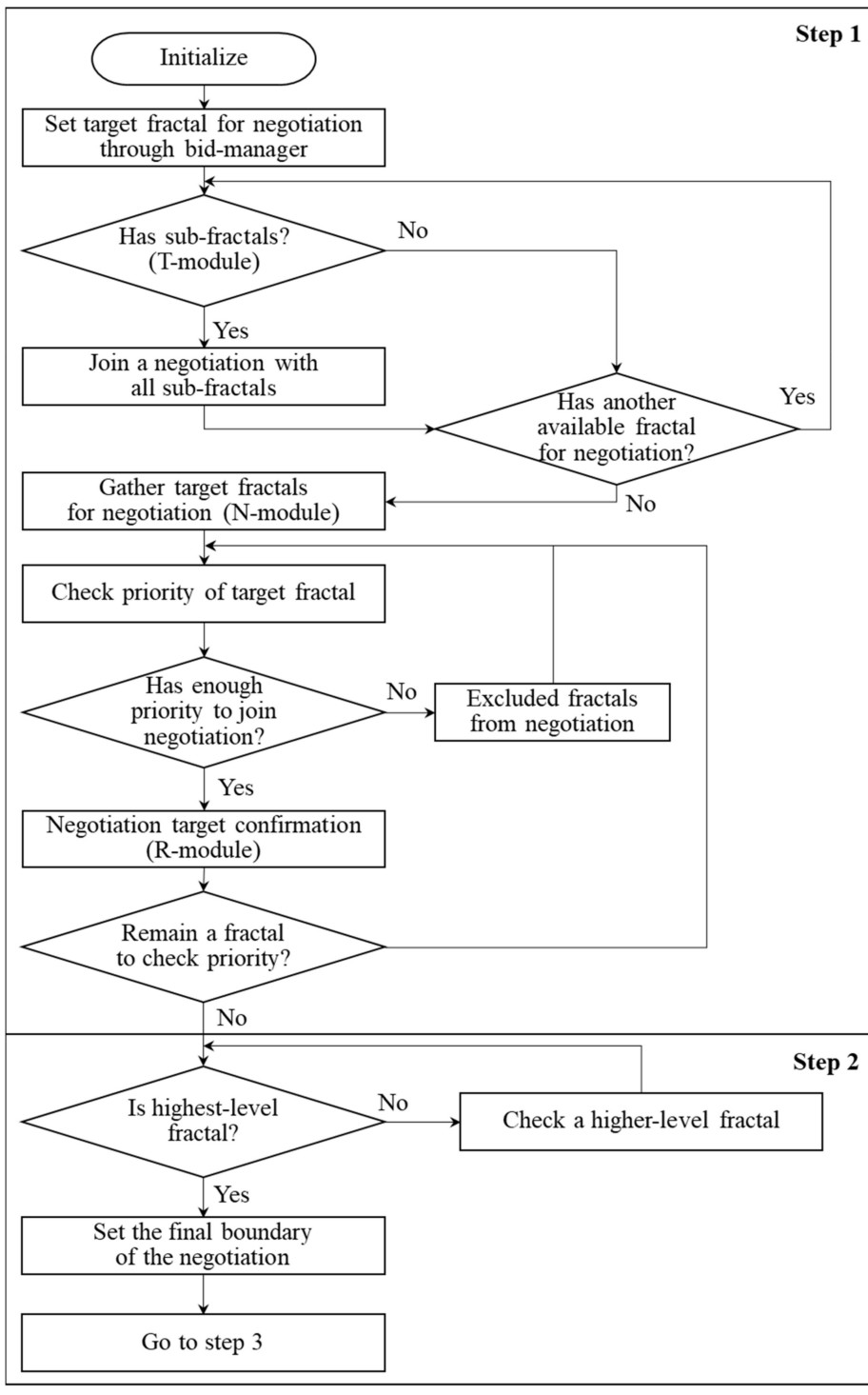

**Figure 17.** Steps 1 and 2 of hybrid-type negotiation.

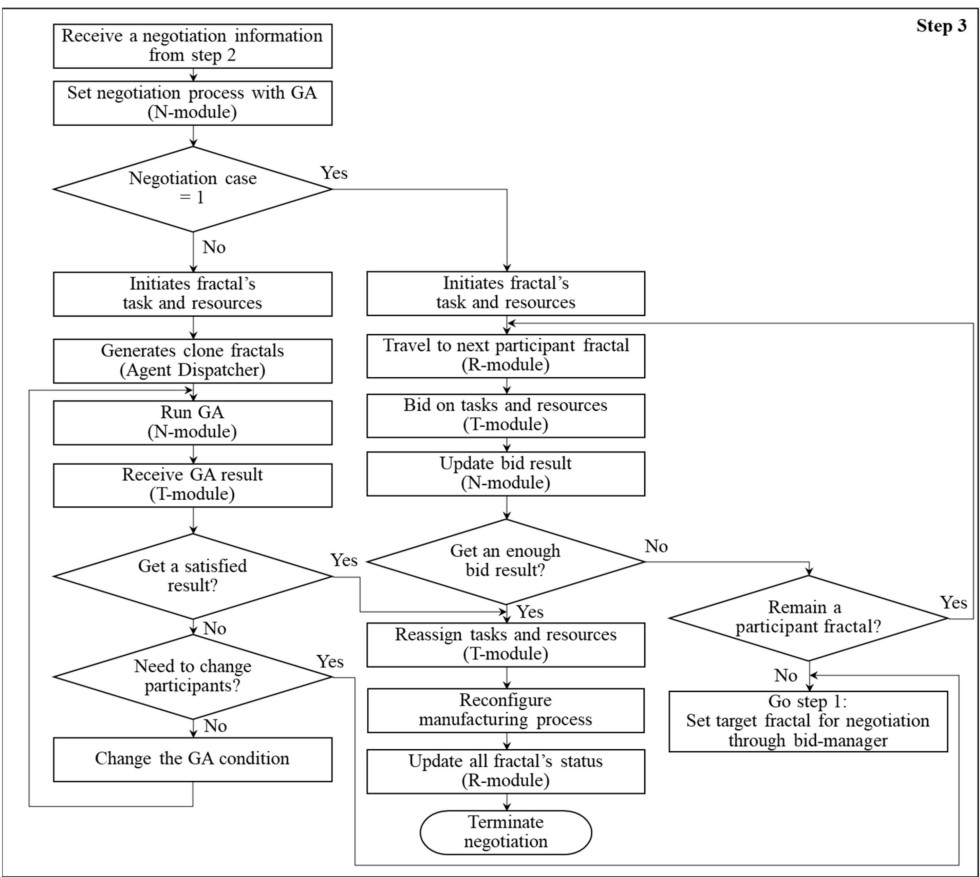

**Figure 18.** Step 3 of the hybrid-type negotiation process.

**Table 4.** Experiment result of GA for negotiation model.

| | | | | | | *(unit: min)* |
| --- | --- | --- | --- | --- | --- | --- |
| | | $F_1$ | $F_2$ | $F_3$ | $F_4$ | $F_5$ |
| Job 1 | $T_{11}$ | 2 | 5 | 4 | 1 | 2 |
| | $T_{12}$ | 5 | 4 | 5 | 7 | 5 |
| | $T_{13}$ | 4 | 5 | 5 | 4 | 5 |
| Job 2 | $T_{21}$ | 2 | 5 | 4 | 7 | 8 |
| | $T_{22}$ | 5 | 6 | 9 | 8 | 5 |
| | $T_{23}$ | 4 | 5 | 4 | 54 | 5 |
| Job 3 | $T_{31}$ | 9 | 8 | 6 | 7 | 9 |
| | $T_{32}$ | 6 | 1 | 2 | 5 | 4 |
| | $T_{33}$ | 2 | 5 | 4 | 2 | 4 |
| | $T_{34}$ | 4 | 5 | 2 | 1 | 5 |
| Job 4 | $T_{41}$ | 1 | 5 | 2 | 4 | 12 |
| | $T_{42}$ | 5 | 1 | 2 | 1 | 2 |

### 5.3. Sustainability Assessment Method

Both sustainability and each fractal's goal are considered together by the sustainability assessment method in SSrMS. The proposed method involves conducting an integrated assessment of system optimization with respect to sustainability. As shown in Figure 19, the analytic hierarchy process (AHP), Green-BOM, and neural network are utilized in the proposed method. Sustainability KPIs and their weights are derived by the AHP based on the determined goal of each fractal. To obtain the new goal, the obtained sustainability KPIs considering each fractal's goal are derived by a neural network with the goal decision model. Furthermore, environmental regulations and other process data are provided by

the Green-BOM. The new goal obtained via the proposed model is then simulated in digital facilities to verify the new system optimization with sustainability.

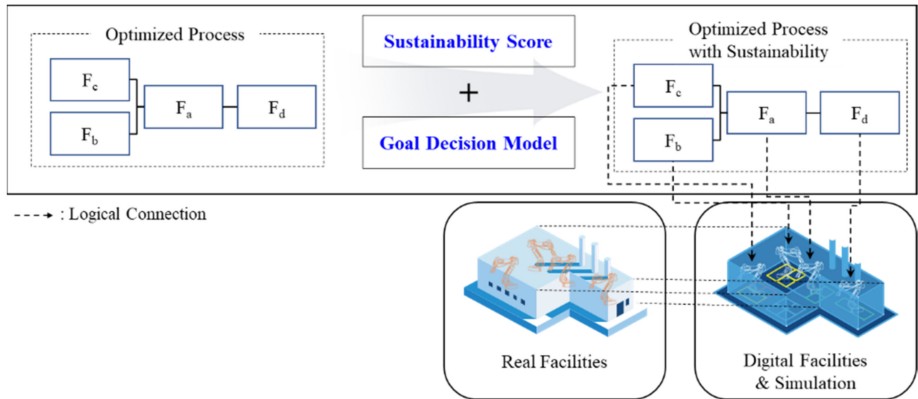

**Figure 19.** Sustainability assessment method in SSrMS.

The overall procedure for the sustainability assessment is shown in Figure 20. The sustainability KPIs are selected based on the global reporting initiative (GRI) framework [52,53] or the demand of the system operator. Based on these selected sustainability KPIs, the system operator implements the AHP to derive the weight of each KPI. Then by BSC, a sustainability score is then calculated based on the weights from the AHP.

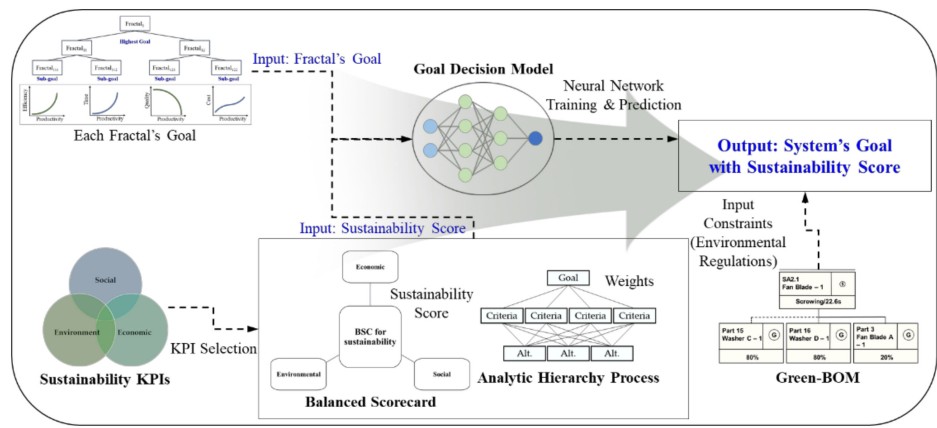

**Figure 20.** Flow of process reconfiguration and optimization by three methods.

In order to derive the sustainability score, the BSC method consists of three perspectives, including the sustainability dimensions, performance measures, weights from AHP result, actual performance, standard performance, performance level, and score as shown in Table 5. Each column in Table 5 can be described as follows:

**Table 5.** BSC for sustainability measurement (modified from [54]).

| Perspective/ Sustainability Dimension | Performance Measures | Weights | Actual Performance | Standard Performance | Performance Level | Score |
|---|---|---|---|---|---|---|
| Economic | | 32.90 * | | | | |
| | Cost | 36.78 | 50 | 20 | 90 | 10.89 |
| | Market presence | 63.22 | 20 | 10 | 90 | 18.72 |
| Environmental | | 44.00 * | | | | |
| | Materials | 30.45 | 95 | 85 | 90 | 12.06 |
| | Energy | 9.55 | 98 | 97 | 60 | 2.52 |
| | Water | 13.86 | 80 | 78 | 60 | 3.66 |
| | Emissions | 14.77 | 10 | 10 | 60 | 3.90 |
| | Effluents and waste | 10.23 | 75 | 76 | 60 | 2.70 |
| | Products and services | 21.14 | 90 | 60 | 90 | 8.37 |

**Table 5.** *Cont.*

| Perspective/ Sustainability Dimension | Performance Measures | Weights | Actual Performance | Standard Performance | Performance Level | Score |
|---|---|---|---|---|---|---|
| | | **23.20 \*** | | | | |
| | Employment | 23.28 | 9 | 12 | 30 | 1.62 |
| | Labor/management relations | 8.62 | 85 | 45 | 90 | 1.80 |
| Social | Training and education | 9.91 | 24.8 | 15 | 90 | 2.07 |
| | Anti-corruption | 34.05 | 98 | 97 | 60 | 4.74 |
| | Public policy | 9.05 | 75 | 30 | 90 | 3.58 |
| | Marketing communication | 15.09 | 25 | 10 | 90 | 5.97 |
| | | | | | Total Sustainability Score | 73.05 |

\* Total weight of each perspective.

In the first column, perspective/sustainability dimensions represent the sustainability dimensions, including the economic, environmental, and social dimensions.

In the second column, the performance measures contain the sustainability KPIs from the GRI framework, which are determined by the system operator.

In the third column, weights are allocated for each performance measure. The bold-faced italicized letters in the column denote the total weight of each perspective. The total weight of each perspective corresponds to the sum of the weights of the sustainability KPIs obtained from the AHP result in performance measures. The weights of each performance measures in weights is a proportional weights of perspective.

The fourth column contains the actual performance of the company or manufacturing system.

The fifth column contains the standard performance, which expresses the average performance of the company or manufacturing system. The average performance refers to the standard level of certain measurements, including energy consumption and emissions in air and water.

The sixth column contains the performance level, for which the actual performance is compared with the standard performance. The performance level includes three categories: excellent, normal, and poor. Excellent indicates that the actual performance exceeds the standard performance, and the value corresponds to 90. Good performance occurs when the two performances are equal or at the same level, and the value corresponds to 60. Bad performance indicates that the standard performance exceeds the actual performance (i.e., the performance level is as usual), and the value corresponds to 30.

The seventh column contains the sustainability score of each performance measure and the total sustainability score of the company or system. This measure is equal to the product of the proportional weight of the measure, the proportional weight of its perspective, and its performance level. For example, the result of economic performance (cost) is 32.90% × 36.78% × 90 = 10.89 (weight of economic dimension × weight of economic performance (cost) × performance level = score).

The sustainability assessment method uses the neural network to assess sustainability with respect to the system's goal optimization using the goal decision model. The input data for the experiment correspond to the lower-fractal's goal and include productivity, efficiency, utilization, flexibility, cost, sustainability score, and profit. A total of 500 data were used in the experiment, and there were no missing values in the data. Prior to the experiment, all the data were preprocessed and normalized. Python 3.8 is used with Jupiter notebook as a tool for the neural network implementation. Training and validation data size is 50%. A total of 500 training epochs were used. The neural network structure consists of one input layer with six nodes, one hidden layer with twenty-four nodes, and one output layer with six nodes. It was assumed that the system's goal corresponds to maximizing profit. The neural network employs the Gaussian function as an activation function, because, based on the data tendency and neural network optimization, the Matlab learner app recommends a Gaussian regression model. After neural network training, the neural network loss rate and MSE value were close to 0.406. To obtain the result from the neural network, a few constraints were assumed: the sustainability score must exceed

0.51, utilization and cost must exceed 0.3, cost is fixed, and usage of resource utilization is minimal. A sustainability score lower than 0.51 is insufficient for sustainable development. The initial results of the neural network are listed in Table 6. Based on these results, it can be concluded that the environmental dimension reduces utilization and increases productivity via labor resource and the social dimension.

**Table 6.** Final goal value of the sustainability assessment method via goal decision model.

| KPIs | Profit | Productivity | Utilization | Flexibility | Cost | Efficiency | Sustainability |
|------|--------|--------------|-------------|-------------|------|------------|----------------|
| Value | 6.77 | 0.82 | 0.3 | 0.3 | 0.3 | 0.38 | 0.51 |

### 5.4. Reconfiguration Flows in SSrMS Architecture

Figure 21 shows the proposed architecture and methods for SSrMS. The self-reconfiguration process starts from equipment at left-bottom in Figure 21. In order to set a new goal and initial manufacturing processes in SSrMS, the goal decision model adopted determines the goals for machines, workstations, and the entire system. In order to optimize manufacturing processes, reconfiguration of such processes is conducted by using the negotiation model equipped with GA. Additionally, sustainability assessment with the goal decision model was implemented via the sustainability assessment method. To realize the system's goal, an individual fractal decides its own goal based on the system's goal. Manufacturing processes are reconfigured by using the negotiation model under scenarios involving addition/removal of fractals in the system or reallocation of tasks/resources. This negotiation model can lead to process optimization. Furthermore, the sustainability of the system is considered, where the sustainability assessment method is used to simultaneously derive sustainability and fractal goals based on the goal decision model. The proposed assessment model employs the AHP to determine sustainability-related KPIs and the Green-BOM to account for environmental regulations.

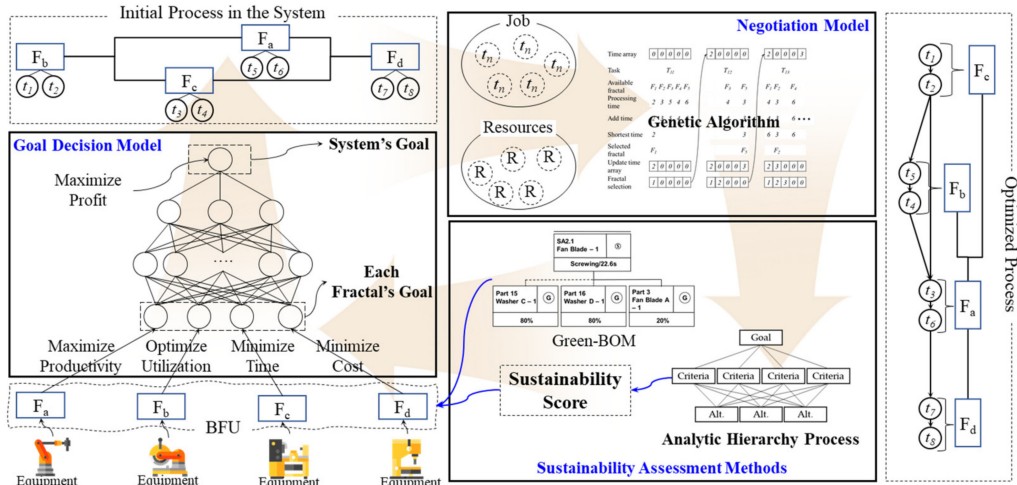

**Figure 21.** Process reconfiguration with the proposed methods in SSrMS architecture.

## 6. Conclusions

This paper proposed the architecture of SSrMS, which is a fully self-reconfiguration-oriented system. A smart, self-reconfigurable manufacturing system (SSrMS) is an improved system with the SMSs concept based on the features of FrMS. SSrMS aims to design architecture and methods for complete autonomous and internal optimization via self-reconfiguration, goal decision, and negotiation between fractals. To realize autonomous functions for the system, self-control methods are required. Thus, the development of autonomous functions and architecture for autonomous operation is essential. The proposed SSrMS is suitable for engineering-to-order manufacturing operations, such as injection

molding, ship manufacturing, aircraft manufacturing, etc., because these types of operations are more compatible with process optimization via process reconfiguration, as opposed to the removal and adjustment of equipment.

The contributions of this paper include the design of a new architecture and the development of autonomous technologies to cope with the paradigm shift caused by the fourth industrial revolution. The proposed SSrMS exhibits the fractal concept and structure of the FrMS, given that the concept and theoretical background of the FrMS is closest to those of SMSs. Furthermore, the FrMS characteristics were optimized for the system as well as for process self-reconfiguration. Therefore, the SSrMS architecture and methods are orientated toward autonomous operations for self-reconfiguration. It is expected that the proposed SSrMS, which has scalable features, can facilitate fundamental studies on SMSs and fully autonomous systems. The proposed SSrMS architecture represents a significant design for the autonomous operations required in SMSs. The self-reconfiguration method in SMSs is afforded by the scalable fractal structure and the various autonomous functions. The goal decision model signifies a method that considers the system's goal and the goal of each piece of equipment. Therefore, the goal decision model can constitute a fundamental approach to simultaneously realize the local and global optima for SMSs. Additionally, it contributes toward deriving the system's goal via an NN based on the complex relationship between goals. The contribution of the negotiation method involves removing the partial inefficiencies in the system using the GA. It imparts homeostasis to the system without a reconfiguration of the entire process. The sustainability assessment corresponds with a significant design to simultaneously consider sustainability and the system's goal. Moreover, the proposed model considers the operator's demand, environmental regulations, and the system's goal via the AHP, Green-BOM, and NN, respectively.

The proposed SSrMS architecture and methods for autonomous functions represent the implement of smartization for autonomous operation. Nevertheless, it is beyond the scope of this study to cover all the related aspects and mechanisms. Furthermore, all of the data for the experiment is modified from the real data. Therefore, three methods still remain for the research to improve the method with real data. It is, therefore, expected that future research can help improve autonomous functions, methods, and mechanisms for SMSs. Thus, further studies should explore the following areas:

First, the development of big data and environmental information systems should be explored. The development of big data is essential for the implementation of SMSs, because SSrMS and SMSs cannot be implemented without data acquisition and utilization. The environmental information system, data collection, preprocessing, and application methods are critical technologies for the self-decision making required to implement autonomous functions. Second, the mechanism of the goal decision model necessitates improvements based on actual data. The experiment results indicate the feasibility of the method, although they exhibit limitations in terms of the implementation of the actual model and the completion level of the NN. Third, the development of a self-learning module is required to guarantee the accuracy of autonomous decision-making. Self-learning is an essential technology for realizing the autonomous operation of SSrMS and SMSs considering environmental changes and system optimization.

**Author Contributions:** Conceptualization, S.L. and K.R.; methodology, S.L.; software, S.L.; validation, S.L. and K.R.; formal analysis, S.L.; investigation, K.R.; resources, S.L. and K.R.; data curation, S.L.; writing—original draft preparation, S.L.; writing—review and editing, S.L. and K.R.; visualization, S.L.; supervision, K.R.; project administration, K.R.; funding acquisition, K.R. All authors have read and agreed to the published version of the manuscript.

**Funding:** This research received no external funding.

**Institutional Review Board Statement:** Not applicable.

**Informed Consent Statement:** Not applicable.

**Data Availability Statement:** Please contact corresponding author. The data used in this paper are hypothetical data, and private enterprise data.

**Acknowledgments:** This work was supported by the National Research Foundation of Korea (NRF) grant funded by the Korea government (Ministry of Science and ICT) (No. 2021R1A2C2009984).

**Conflicts of Interest:** The authors declare no conflict of interest.

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
