# Peer review of "Development of the Architecture and Reconfiguration Methods for the Smart, Self-Reconfigurable Manufacturing System"

_applsci, doi:10.3390/app12105172_

Round 1

Reviewer 1 Report

I assume that Authors have neglected to erase the sentence (line 8-9). The paper needs to be corrected not only by English native speaker but also in term of syntax composition and grammar. In the line 11 should be stated Manufacturing. Capital letter after full stop. Line 23 please clarify the sentence. The sentence beginning with “The proposed SSrMS” in the line 27 should also be reconstructed in order to be clearer and more precise. I suggest that Authors use the same tense for describing the model (line 102-103 and then 106-107).

Very interesting approach analysing a smart self-reconfigurable manufacturing system that is based on the fractal manufacturing system. Introducing the proposed system architecture is clearly new and the novelty should be highlighted.

Please provide in abstract the main findings of your study. It is not clear what is analysed and what are the results.

In the end of the Introduction the structure of the paper should be given.

The Literal review should include new references tackling the same challenges and used methods.

I consider that Methodology of the research should be clearer and more specific. What is the scientific contribution of the analysis and achieved results?

How does the proposed methodology used in this research differ from other methods and what are the advantages? Please indicate what are the weaknesses of the proposed methodology.

The last sentence in the Discussion should be more highlighted and better determined. I found the discussion part poorly written. It should be clearer and more concise. The discussion part should relate to the results of other authors.

The limitation of study should be addressed, and future research guidelines need to be provided in conclusions.

Figure 17 should be corrected with capital letters.

Author Response

Please refer to the file contents attached.

Reviewer 2 Report

The article is devoted to developing an intelligent self-reconfiguring production system based on a fractal production system. The study's relevance is dictated by the fact that over the past decades, the demand for smarter and more intelligent production systems has increased to meet customers' growing needs. Intelligent manufacturing systems are capable of fully integrated autonomous operation. The concept of autonomous systems and functions is adopted for following generation production systems. The architecture of an intelligent self-reconfiguring production system proposed by the authors is designed to implement self-reconfiguration functions based on the concept of a fractal production system. The intelligent self-reconfiguring production system has a fractal structure, which allows the distribution of control functions. It also forms the fundamental basis for autonomous operation and reconfiguration between each fractal. The authors propose three reconfiguration methods for system reconfiguration: a goal decision model, a negotiation model, and a sustainability assessment method. The Goal Decision Model is designed to determine the goal of each fractal and system. The negotiation model is adopted to perform partial optimization of the process by redistributing tasks and resources between fractals, based on the goal of coping with changes in the state of the system. The sustainability assessment method is designed to simultaneously assess sustainability concerning the system's goals. The proposed intelligent self-reconfiguring manufacturing system architecture and three specialized methods for self-optimization, self-organization, and self-reconfiguration.

Despite the satisfactory quality of the article, some shortcomings need to be corrected.

  1. The aim of the article should be defined.
  2. Figures 9 and 10 should be redrawn. The text is superimposed on the drawing, making it difficult to interpret.
  3. It is recommended to describe each step of the reconfiguration flow in more detail.
  4. The state-of-the-art methods should be separated from the ones proposed by the authors.
  5. The architecture of the neural network should be described in detail.
  6. The data for the experimental study should be described.
  7. The numerical results obtained by the authors should be discussed.
  8. The scientific novelty of the paper should be highlighted.
  9. It is recommended to include more methods considering sustainability assessment in research analysis. Such methods do not have to be based on neural networks, e.g. doi: 10.3390/en14248235

In summarizing my comments, I recommend that the manuscript is accepted after major revision. 

Author Response

(The authors gave the same response as above.)

Round 2

Reviewer 1 Report

Dear Authors,

Thank you for including all my remarks. Now the scientific soundness is on the high level. Thank you for your effort.

Reviewer 2 Report

Thanks for the authors for considering all the comments and recommendations. In my opinion, now the paper can be accepted.